# Acute Downregulation but Not Genetic Ablation of Murine MCU Impairs Suppressive Capacity of Regulatory CD4 T Cells

**DOI:** 10.3390/ijms24097772

**Published:** 2023-04-24

**Authors:** Priska Jost, Franziska Klein, Benjamin Brand, Vanessa Wahl, Amanda Wyatt, Daniela Yildiz, Ulrich Boehm, Barbara A. Niemeyer, Martin Vaeth, Dalia Alansary

**Affiliations:** 1Molecular Biophysics, Saarland University, 66421 Homburg, Germany; priska.jost@uks.eu (P.J.); s8faklei@stud.uni-saarland.de (F.K.); barbara.niemeyer@uks.eu (B.A.N.); 2Würzburg Institute of Systems Immunology, Max Planck Research Group at Julius-Maximilians University of Würzburg, 97078 Würzburg, Germanymartin.vaeth1@uni-wuerzburg.de (M.V.); 3Experimental Pharmacology, Center for Molecular Signaling (PZMS), School of Medicine, Saarland University, 66421 Homburg, Germany; vanessa.wahl@uks.eu (V.W.); amanda.wyatt@uks.eu (A.W.); daniela.yildiz@uks.eu (D.Y.); ulrich.boehm@uks.eu (U.B.)

**Keywords:** mitochondrial calcium uniporter, regulatory T cells, suppressive capacity

## Abstract

By virtue of mitochondrial control of energy production, reactive oxygen species (ROS) generation, and maintenance of Ca^2+^ homeostasis, mitochondria play an essential role in modulating T cell function. The mitochondrial Ca^2+^ uniporter (MCU) is the pore-forming unit in the main protein complex mediating mitochondrial Ca^2+^ uptake. Recently, MCU has been shown to modulate Ca^2+^ signals at subcellular organellar interfaces, thus fine-tuning NFAT translocation and T cell activation. The mechanisms underlying this modulation and whether MCU has additional T cell subpopulation-specific effects remain elusive. However, mice with germline or tissue-specific ablation of *Mcu* did not show impaired T cell responses in vitro or in vivo, indicating that ‘chronic’ loss of MCU can be functionally compensated in lymphocytes. The current work aimed to specifically investigate whether and how MCU influences the suppressive potential of regulatory CD4 T cells (Treg). We show that, in contrast to genetic ablation, acute siRNA-mediated downregulation of *Mcu* in murine Tregs results in a significant reduction both in mitochondrial Ca^2+^ uptake and in the suppressive capacity of Tregs, while the ratios of Treg subpopulations and the expression of hallmark transcription factors were not affected. These findings suggest that permanent genetic inactivation of MCU may result in compensatory adaptive mechanisms, masking the effects on the suppressive capacity of Tregs.

## 1. Introduction

Mitochondria represent a cellular hub where intracellular signals converge then become translated into proliferation, differentiation, or apoptotic processes. Mitochondria, therefore, play a particularly crucial role in T cell activity. T cell activation triggers remodeling and polarization of mitochondria close to the immunological synapse to secure ATP levels, meeting the increasing demands of activated cells, and to buffer Ca^2+^ away and guard against Ca^2+^-dependent channel inactivation, thus sustaining Ca^2+^ influx [1]. Furthermore, mitochondria support T cell differentiation by inducing reactive oxygen species (ROS) essential for the nuclear factor of activated T cells (NFAT), nuclear factor k-light-chain-enhancer of activated B cells (NF-kB), and activator protein-1 (AP-1) activity and for metabolic reprogramming to change the dependence of quiescent naïve T cells on oxidative phosphorylation to the more efficient energy productive glycolytic metabolism, in a process termed the Warburg effect [2,3,4].

Being a central organelle, mitochondria have their own machineries for Ca^2+^ homeostasis. In addition to the outer mitochondrial-membrane-resident voltage-dependent Ca^2+^ channels, the mitochondrial Ca^2+^ uniporter (MCU) represents a major pathway for mitochondrial Ca^2+^ uptake. MCU is a highly selective Ca^2+^-conducting channel that assembles together with the non-conducting MCUb and the MCU-regulatory (EMRE) subunits to assemble the MCU protein complex resident in the inner mitochondrial membrane. The activity of MCU is crucially regulated by two EF-hand-containing subunits (MICU1 and MICU2) that allow opening of MCU only upon a detectable rise in cytosolic Ca^2+^ (reviewed in [5]). Genetic deletion of *Mcu* in different mouse models did not result in overt survival phenotypes, while a conditional deletion model was protective against hypoxic/ischemic brain injury [6]. Corresponding analysis of the role of MCU in T cells was mainly lacking until 2021, when Yoast and colleagues showed that downregulation of MCU in a T cell line results in an accumulation of cytosolic Ca^2+^ and enhanced NFAT translocation [7]. This timely study, however, did not entail investigation of functional consequences of MCU downregulation in T cells, which was investigated systematically in a parallel study [8]. Using a mouse model that specifically deletes *Mcu* in CD4 T cells, Wu and colleagues showed that MCU is largely dispensable for murine T cells function [8]. In agreement with Yoast and colleagues, *Mcu* knock-out resulted in elevated cytosolic Ca^2+^ in response to thapsigargine (Tg)-mediated store depletion. The elevated cytosolic Ca^2+^, however, failed to enhance T cell proliferation, differentiation, or effector function. Furthermore, mitochondrial respiratory profiles of T cell effector subpopulations remained unchanged despite ablation of MCU.

Regulatory (Treg) cells are a unique CD4^+^ T cell population able to terminate or dampen immune responses and are, thus, important for maintaining self-tolerance. The Treg specific hallmark transcription factor Forkhead-Box-Protein 3 (FoxP3) sustains expression of IL-2 receptor CD25 to favour Treg survival over that of CD4 conventional cells (Tcons) [9]. Furthermore, Tregs can suppress Tcons by direct contact and inhibition of Ca^2+^-mediated T cell activation [10] or indirectly by release of suppressive cytokines, mainly tumor growth factor-β (TGF-β), interleukin-10 (IL-10), and IL-35, or by inhibition of dendritic-cell-mediated T cell activation [11]. Dysfunctionality of Tregs is, therefore, a common dominator of chronic inflammatory and autoimmune diseases. Conversely, the suppressive nature of Tregs and their potential for persistence as memory cells make them promising candidates for the therapeutic manipulations, including adaptive Treg transfer and Treg-based chimeric antigen receptor (CAR) therapies [12]. Metabolically, Tregs rely predominantly on oxidative phosphorylation and only modestly on glycolysis, making regulation of mitochondrial function of particular importance for Treg function. Recent studies showed that within the course of an autoimmune disease model, Tregs exhibit an elevated oxidative stress response, attenuation of lysosomal function, and ultimately cell death, recapitulating the metabolic reprogramming evident in peripheral Tregs from patients with autoimmune diseases [13]. This metabolic reprogramming is largely dependent on mitochondrial Ca^2+^ homeostasis [5]. 

In the previous study by Vaeth group, *Mcu* was specifically targeted in Tregs by using a *Foxp3^Cre^ Mcu^fl/fl^* mouse model [8]. Although this approach did not result in Treg phenotypic alterations, and mice showed comparable susceptibility to infection, the question remained whether long-term adaptive mechanisms could be masking a functional Treg-specific effect. To address this question, we set out in the current study to compare the effects of acute siRNA-mediated downregulation with genetic ablation of *Mcu* in Tregs.

## 2. Results

### 2.1. siMcu Treatment Sustains Low Protein Levels during the Course of Suppression Assay

Before comparing the effects of genetic ablation with the transient downregulation of *Mcu* on suppressive capacity, we first analyzed the efficiency of transient downregulation of *Mcu*. To this end, we transfected murine Tregs with *Mcu*-specific siRNA. Analysis of *Mcu*-mRNA 72 h after transfection showed that treatment with si*Mcu* resulted in reduction in *Mcu* mRNA levels, down to 11.5 ± 1.4% of corresponding levels in control cells transfected with non-targeting RNA (NT) (Figure 1A). Because our aim was to measure the suppressive capacity of Tregs after downregulation of *Mcu* in the suppression assay that entails TCR stimulation, we extended the monitoring of the effect of siRNA transfection over 6 days with or without stimulation. Analyses of protein levels showed that extending the cell culture for 6 days after transfection with or without stimulation with anti-CD3/CD28-coated beads ameliorated the siRNA-mediated downregulation observed on Day 3 (down to 21 ± 4%) and resulted in partial recovery so that we now measure 38 ± 4% without stimulation and 44 ± 7% with bead stimulation (Figure 1B,C).

### 2.2. Downregulation of Mcu Impairs Mitochondrial but Not Cytosolic Ca^2+^ Uptake

Because MCU constitutes the pore-forming unit in the main mitochondrial Ca^2+^ uptake pathway, we investigated the effect of *Mcu* downregulation on Ca^2+^ homeostasis. First, we transduced murine T cells with adenovirus-encoding Ca^2+^ sensor targeted to the mitochondria (mtCEPIA3). Measurements performed using two independent murine T cell lines—EG7 (Figure 2A) and EL-4 (Appendix A)—show that si*Mcu* resulted in significant reduction of mitochondrial Ca^2+^ uptake induced by ATP (Figure 2B and Appendix A) or by histamine (Figure 2C and Appendix A) application, in agreement with the efficient downregulation we observed (Figure 1) and with the previous findings using murine T cells with CD4-specific deletion of *Mcu* (*Mcu^fl/fl^Cd4^Cre^*) [8].

In Wu et al., it was shown that genetic ablation of *Mcu* enhances store-operated Ca^2+^ entry (SOCE) in response to thapsigargine (Tg) application, which bypasses TCR stimulation [8]. Therefore, we asked whether altering MCU expression levels equally enhances TCR-activation-induced SOCE in a CD4 subpopulation differential manner. While Tcons recapitulated the enhancement of Tg-induced SOCE following siRNA-mediated downregulation of *Mcu* (Figure 3A,B), genetic ablation did not alter TCR-activation-induced SOCE in Tcons (Figure 3C). Interestingly, TCR-stimulation-induced SOCE in Tregs was affected neither by genetic ablation nor by siRNA-mediated downregulation of *Mcu* (Figure 3D,E). These results indicate that MCU plays a differential role in T cell subpopulations, which might be masked by adaptive mechanisms associated with genetic ablation.

### 2.3. Mcu Is Dispensable for T Cell Fate Decision

Although the previous study by Wu and colleagues demonstrated that genetic ablation of *Mcu* had no effect on proliferation or differentiation of CD4 T cells into effector subpopulations, we asked whether siRNA treatment would unravel effects masked by adaptive mechanisms. To address this question, we isolated CD4 T cells from WT mice and subjected them to non-targeting (NT) or si*Mcu* treatment then monitored the distribution of cells into naïve (N), effector memory (EM), or central memory (CM) compartments based on CD44 and CD62L surface expression with and without stimulation with anti-CD3/CD28-coated beads. Representative plots of the flow cytometric analysis in Figure 4A–D show exemplary compartment distribution of Tregs (Figure 4A) or of Tcons after isolation (Figure 4B) or 3 days after transfection with non-targeting RNA (Figure 4C) and on Day 6 following transfection, while cells have been stimulated on Day 3 with anti-CD3/CD28 beads (Figure 4D). Quantification of the fractions of N, EM, or CM compartments shows that Tcons and Tregs exhibit comparable distribution (Figure 4E). Furthermore, compartment analysis at Day 6 after transfection shows that stimulation of Tcons with anti-CD3/CD28 beads shifts the distribution towards higher EM fraction. However, the compartment distribution was not altered by siRNA treatment, indicating that in agreement with the genetic ablation, MCU is dispensable for CD4 T cell differentiation and fate decision.

### 2.4. Downregulation of Mcu Reduces Suppressive Capacity of Tregs

Treg-specific genetic ablation of *Mcu* (*Mcu^fl/fl^Foxp3^Cre^* mice) did not lead to an overt spontaneous immune reaction in mice not subjected to an additional disease model nor did challenging the *Mcu^fl/fl^CD4^Cre^* mice in experimental autoimmune encephalomyelitis (EAE) or lymphocytic choriomeningitis virus (LCMV) infection models [8], indicating that MCU is dispensable for function of Tregs, but permanent MCU ablation may be compensated by other Ca^2+^ influx mechanisms in vivo. However, in this previous study, the *Mcu^fl/fl^Foxp3^Cre^* mice were not challenged; here, we aimed to directly test whether MCU regulates suppressive capacity of Tregs in vitro. To this end, we measured the proliferation of WT CD4 Tcons while co-cultured with either Tregs from *Mcu^fl/fl^Cd4^Cre^* or *Cd4^Cre^* mice or, alternatively, with WT Tregs treated with NT or si*Mcu* during anti-CD3/CD28 stimulation. Comparison of the suppressive capacity in both experimental designs shows that while genetic ablation did not affect in vitro suppressive capacity of Tregs (Figure 5A), siRNA-mediated downregulation of *Mcu* led to a consistent and significant reduction of the suppressive capacity to 32.5 ± 7% (*n* = 6) compared with the NT-transfected cells (Figure 5B). The reduction in suppressive capacity was not underlain by alteration in transcription of hallmark transcriptional factors FoxP3 or HELIOS in Tregs (Figure 5C). Because the suppressive capacity also relies on production of suppressive cytokines, such as IL-10, IL-35 (a heterodimer of IL-12A and Epstein–Barr-virus-induced gene 3 (Ebi-3)) and TGF-β [14,15], we also measured the corresponding expression levels in Tregs at Day 3 after transfection. Indeed, we observe a reduction, albeit not significant (*p* = 0.1), in expression of IL-10 (Figure 5D, left panel). Although we observe an increase in expression of Ebi-3, conclusions about levels of IL-35 are hampered since we were unable to detect expression of IL-12A, essential for heterodimerization with Ebi-3 and assembly of IL-35 [14]. In contrast to IL-10, levels of TGF-β in si*Mcu*-transfected Tregs were comparable to control non-targeting RNA-transfected cells.

## 3. Discussion

Perturbation of genes and generation of knockout mice have been established as an indispensable tool in immunological studies ever since their initial application in studying MHC function in 1990 [16,17]. The utilization of restriction tools, such as the cre-loxP system, not only overcame potential lethality of global knockouts but has also added significant versatility to knockout strategies by allowing conditional or cell-type-specific gene deletion [18]. Despite the powerfulness of knockout strategies and the ability to spatially and temporally control gene deletion, knockouts suffer the intrinsic caveat of the genetic robustness or canalization [19]. The phenomenon of genetic robustness entails the ability of organisms to alter metabolic, signaling, or transcriptional networks or even introduce mutations in redundant genes to compensate for the deleted genes and may result in incomplete or paradoxical results [20,21]. In comparison with permanent gene deletion, siRNA-mediated downregulation is associated with less financial and work time burden and were, thus, considered to provide an alternative to avoid compensatory mechanisms. Furthermore, the siRNA approach does not depend on inducing frame shift mutations or ploidy, thus avoiding or minimizing genetic heterogeneity [22]. Using siRNAs, however, often leads to off-target effects and comes with the caveat of possibly mediating hypomorphic phenotypes due to incomplete loss of function in addition to reversibility, although this can be useful in some studies [22,23]. 

In the current study, we compared the effects of acute siRNA-mediated downregulation with genetic ablation of *Mcu* in CD4 T cells (from *Mcu^fl/fl^Cd4^Cre^* mice) on suppressive capacity of Tregs. Wu and colleagues previously showed that genetic ablation of *Mcu* failed to alter adaptive immunity in either an autoimmune (EAE) or an infection (LCMV) model, indicating that the suppressive capacity was not altered in vivo [8]. Because these experiments used *Mcu^fl/fl^Cd4^Cre^* mice and, therefore, targeted MCU in both Tcon and Treg populations, it was important to investigate whether a Treg-specific effect was masked in this experimental design. Wu and colleagues addressed this question by generating a mouse line with Treg specific deletion of *Mcu* (*Mcu^fl/fl^Foxp3^Cre^* mice), which turned out to be comparable to the control littermates concerning T cell development and numbers of peripheral Treg. Furthermore, Treg-specific deletion of *Mcu* did not result in discernable immune phenotype and, thus, did not challenge the mice in a disease model [8]. Nevertheless, since an absolute certainty about the interpretation of the results from the knockout models is hampered by the aforementioned caveats of adaptive mechanisms, we utilized an acute approach for downregulation in the current study. Indeed, we found that acute siRNA-mediated downregulation of *Mcu* resulted in moderate but consistent inhibition of suppressive capacity. We also propose that the reduction we measured is attenuated by instability of the siRNA effect and the concomitant recovery of MCU levels during the course of the in vitro suppression assay (Figure 1). The decreased suppressive capacity was not associated with a parallel reduction of FoxP3 (Figure 5C). This finding is reminiscent of the loss of suppressive function despite a stable expression of FoxP3, Treg survival, and proliferation in a model for Treg-specific deletion of the mitochondrial respiratory complex III [24]. Although we observed a reduction of the suppressive cytokine IL-10, an unambiguous explanation of the difference between the effects of acute and chronic knockdown of *Mcu* could potentially be best tackled by performing a comparative transcriptome and metabolome analysis to discover alterations in calcium homoeostasis and metabolic and redox machineries associated with both approaches and, therefore, presents plausible plan for further work.

Both siRNA and knockout approaches resulted in significant reduction in mitochondrial Ca^2+^ uptake in different cell types (Figure 2 and [7,8,25,26,27]). However, the effect of inhibiting mitochondrial Ca^2+^ uptake on cytosolic Ca^2+^ influx has been the subject of a long-standing debate. The initial notion that SOCE is supported by mitochondrial Ca^2+^ uptake arose from the observation by Hoth and colleagues that pharmacological dissipation of mitochondrial membrane potential and subsequent inhibition of mitochondrial Ca^2+^ uptake ameliorates SOCE in T cells [28], which was reproduced in other cell types [7,29]. This notion was supported in studies utilizing siRNA-targeting *Mcu* [29,30,31,32] but also challenged by the finding that genetic ablation of *Mcu* resulted in increased SOCE (Figure 4 and [7,8,33]). Our current understanding is that the outcome of modulation of SOCE by mitochondrial Ca^2+^ uptake depends largely on cell type and activation state. In addition to conclusions from the aforementioned studies, this hypothesis is augmented by the finding that, while pharmacological inhibition of Jurkat cells (resting human T cell line) inhibited SOCE, parallel treatment of in vitro differentiated primary human primary T cells evoked a significantly enhanced SOCE [34]. Furthermore, T cell differentiation is accompanied by mitochondrial remodeling [35] and by significant alterations of the contribution of PMCA to modulation of SOCE [36]. Importantly, perturbation of the function of MCU results in prominent changes of metabolic and oxidative status of the cells with functional consequences that are difficult or impossible to completely delineate from the effects on Ca^2+^ homeostasis, especially considering other molecular players regulating mitochondrial Ca^2+^ (for comprehensive reviews see [37,38,39]). 

A significant finding of the current study is that in contrast to Tcons, Tregs treated with si*Mcu* did not show an enhancement of TCR-activation-induced SOCE (Figure 2). This finding supports the hypothesis that Ca^2+^ homeostasis in Tregs relies on distinct mechanisms compared with Tcons [34,40]. Furthermore, amelioration of suppressive capacity by si*Mcu* treatment indicates that MCU modulates Treg function with mechanisms largely independent of cytosolic Ca^2+^, a hypothesis that requires further investigation. 

In conclusion, the current study suggests a potential Treg-specific function of MCU and draws attention to the impact of technical approaches on interpretations concerning gene function. A deeper insight into transcriptional alterations consequent to dysregulated expression and/or function of MCU might shed light on disease relevant mechanisms of regulation of T cell function. 

## 4. Materials and Methods

### 4.1. Animal Studies

*Mcu*^fl/fl^ [41] (strain 029817) and Cd4^Cre^ mice were purchased from the Jackson laboratories and were housed at the University of Würzburg. For experiments performed with WT mice, C57BL6/J housed at the University of Saarland were used. All mouse lines were bred under specific pathogen-free conditions and fed with standard chew diet. Housing the animals and organ collection were carried out according to guidelines of local animal protection committee and were approved by local authorities.

### 4.2. Isolation of CD4 T Cells

Mice were euthanized by cervical dislocation, then spleen and lymphocytes were dissected and passed through a 70 µM sieve. Erythrocytes were lysed with buffer containing 155 mM NH4Cl, 10 mM KHCO3, and 0.1 mM EDTA, pH 7.3. Cells were washed twice using Hank’s buffered salt solution (HBSS) containing 2 mM EDTA and 1% penicillin/streptomycin. Cells were then cultured overnight in RPMI164 medium containing 1% penicillin/streptomycin, 50 µM β-mercaptoethanol, and 20 ng/mL IL-2. Next day, Tregs and Tcons were isolated using CD4^+^CD25^+^ Regulatory T Cell Isolation Kit (Miltenyi, #130-091-041) according to manufacturer’s instructions. Alternatively, total CD4 cells were labelled with anti-CD4 PerCP (Biolegend, San Diego, CA, USA, #100431) and isolated using fluorescence-activated cell sorting (FACS-SH800, Sony Biotechnology, CA, USA).

### 4.3. Transfection

For downregulation of *Mcu*, 1 × 10^6^ Tregs or Tcons were transfected with 1 μM siRNA-targeting *Mcu* (Accell Mouse *Mcu* siRNA-SMARTpool, E-062849-00-0010, Dharmacon, Lafayette, CO, USA) or non-silencing control RNA (Accell Non-targeting Pool, D-001910-10-20, Dharmacon, Lafayette, CO, USA) using P3 Primary Cell 4D-Nucleofactor^TM^ Kit (Lonza, Basel, Switzerland) according to manufacturer’s instructions. Tcons were cultured in medium containing 20 ng/mL IL-2 after transfection, and 24 h later, 1 mg/mL PHA-P was included within to improve cell survival. Tregs were cultured in maintenance medium containing 50 ng/mL IL-2, 5 ng/mL TGF-β, and mouse T activator CD3/CD28 Dynabeads at a ratio of 1:10 (beads:cells). Experiments were performed 72 h after transfection.

To monitor the effect of downregulation of *Mcu* on mitochondrial Ca^2+^ uptake, EG7 or EL4 cells were transfected with 1 µM siRNA-targeting *Mcu* using SF Cell Line 4D-Nucleofector^TM^ Kit (Lonza, Basel, Switzerland) according to manufacturer’s instructions. The mitochondrial Ca^2+^ indicator mtCEPIA3 was transduced as described in 4.9. Measurements were performed 48 h after transfection.

### 4.4. Flow Cytometric Analysis

For flow cytometric analysis, cells were washed with phosphate buffered saline (PBS) and stained with a fixable viability dye (Zombie NIR Fixable Viability Kit, Biolegend, San Diego, CA, USA), followed by staining of surface markers. Where indicated, intracellular staining of transcription factors was performed by fixing and permeabilizing the cells using buffers and instructions of commercially available kit (FixPerm # AB_2869008, BD, Franklin Lakes, NJ, USA). Antibodies used were supplied from BioLegend (San Diego, CA., USA) (anti-CD25-PE # 101903, anti-CD4-PerCP # 100431, anti-CD4-BrilliantViolet421 # 100437, anti-CD44-APC # 103012, anti-CD62L-PE/Cy7 # 104418, anti-FoxP3-AlexaFluor647 # 320013, and anti-Helios-PacificBlue # 137210). Flow cytometric analysis was performed with BD FACS Verse^TM^ cytometer and FlowJo Software (BD Life Sciences, Franklin Lakes, NJ, USA).

### 4.5. Suppression Assay

Suppressive capacity of Tregs was measured by analyzing the proliferation of co-cultured Tcons derived from WT during the course of TCR stimulation. To this end, Tcons were stained with 2 µM Carboxyfluorescein succinimidyl ester (CFSE, BioLegend, San Diego, CA, USA) in PBS, and staining was terminated by adding RPMI + 10% FCS. Labelled Tcons were then cocultured with Tregs at a 1:1 ratio while being stimulated with Mouse T Activator CD3/CD28 Dynabeads (3:1 ratio, T cells:Beads, # 11453D, ThermoFischer Scientific, Waltham, MA, USA). After three days of co-culture, Tcon proliferation was analyzed by flow cytometry using BD FACS Verse^TM^ cytometer and FlowJo Software (BD Life Sciences, Franklin Lakes, NJ, USA).

### 4.6. RNA Isolation, cDNA Synthesis, and Quantitative Real-Time PCR (qRT-PCR)

The indicated cell types were harvested and stored at −80 °C until RNA was isolated using Monarch^®^ Total RNA Miniprep Kit (New England Biolabs, Ipswich, MA, USA) following manufacturer’s instructions. SuperScriptTMII Reverse Transcriptase (Life technologies, Carlsbad, CA, USA) was used to generate complementary DNA (cDNA), and subsequent qRT-PCR was conducted using QuantiTect SYBR Green Kit (Qiagen, Hilden, Germany) and a CFX96 Real-Time System (Biorad, Hercules, CA, USA). For quantification, threshold cycle (Cq) values of a gene of interest were normalized to that of TATA box-binding protein (TBP) using the 2^-ΔCq^ method. Primers used were either supplied by Qiagen (QuantiTect Primer Assays, Mm_Mcu_1_SG/QT01065001, Mm_Tbp_1_SG/QT00198443) or synthetized by Eurofins using the following sense and anti-sense sequences, respectively: 5′-CAGAGCCACATGCTCCTAGA-3′ and 5′-TGTCCAGCTGGTCCTTTGTT-3′ for IL-10 [42], 5′-GCTCCCCTGGTTACACTGAA-3′ and 5′-ACGGGATACCGAGAAGCAT-3′ for Ebi-3 [42], 5′-TCAGAATCACAACCATCAGCA-3′ and 5′-CGCCATTATGATTCAGAGACTG-3′ for IL-12A [42], and 5′-ACTGATACGCCTGAGTGGCT-3′ and 5′-CCCTGTATTCCGTCTCCTTG-3′ for TGF-β [43].

### 4.7. Single Cell Ca^2+^ Imaging

For monitoring changes in [Ca^2+^]i, cells were loaded with 1 µM Fura 2-AM in culture medium at room temperature for 30 min with gentle rocking. Stained cells were seeded on (0.1 mg/mL) poly-L-ornithine-coated coverslips. All experiments were at room temperature as in [34]. TCR-activation-induced Ca^2+^ influx was triggered by perfusion of Mouse T Activator CD3/CD28 Dynabeads (ThermoFischer Scientific, Waltham, MA, USA) in an external solution containing 0.5 mM CaCl_2_, 2 mM MgCl_2_, 4 mM KCl, 10 mM Glucose, and 10 mM Hepes. Images were acquired at 0.2 Hz. In addition to fluorescent images, infra-red images were acquired to detect cell–bead contact time points. The area under the influx curve (AUC) was analyzed using Igor and GraphPad prism software. Here, 100–250 cells were measured in 1–2 independent experiments, and 4–5 mice were used per condition. 

### 4.8. Mitochondrial Ca^2+^ Imaging

Cells expressing mtCEPIA3 were starved for 1–2 h in low glucose buffer containing (in mM) 2.5 CaCal_2_, 145 NaCl, 4 KCl, 0.5 Glucose, 10 HEPES, and 0.1% BSA before measurements were started. Cells were then seeded on (0.1 mg/mL) poly-L-ornithine-coated coverslips and permeabilized with 0.04% digitonin in Ca^2+^-free solution containing 1 mM EGTA, 2 mM MgCl_2_, 4 mM KCl, 10 mM Glucose, and 10 mM Hepes. Cells were then washed with intracellular medium according to [5] (ICM contained: 10 mM NaCl, 120 mM KCl, 1 mM KH_2_PO_4_, and 20 mM HEPES). Mitochondrial Ca^2+^ uptake was triggered by application of 0.5 mM Ca^2+^ solution and 100 µM histamine or ATP. For quantification, the area under the influx curve (AUC) was analyzed using Igor and GraphPad prism software. 

### 4.9. Viral Transduction

The sequence encoding mtCEPIA3 was amplified from plasmid #58219 (Addgene, Watertown, MA, USA), with primers adding EcoRI and HindIII restriction sites to subclone the amplicon in AAV-CAG vector (Addgene, Watertown, MA, USA, plasmid #51904), which was modified to expand the multiple cloning site. Using a high-concentration endotoxin-free preparation of this vector, AAV was produced using the triple transfection helper-free method in HEK293T cells. For this, the final vector described above was transfected alongside a vector containing viral genes (pAdDeltaF6 was a gift from James M. Wilson; Addgene, Watertown, MA, USA, plasmid #112867) and a serotype 8-determining vector (pAAV2/8 was a gift from James M. Wilson; Addgene, Watertown, MA, USA, plasmid #112864) in a 1:1:1 ratio. Transfection was undertaken when the cells reached 60–70% confluence using a 4:1 (*v*:*w*) ratio of Polyethylenimine (PEI) to plasmid DNA. Then, 60–72 h after transfection, cells were pelleted and processed to recover the virus; this included isolation and purification of AAV particles from both cells and media using filtration, PEG precipitation steps, and centrifugation. Viral titer was measured by qPCR analysis with primers specific to the ITR region of the packaging plasmid (fwd ITR primer: 5’-GGAACCCCTAGTGATGGAGTT and rev ITR primer: 5’-CGGCCTCAGTGAGCGA). For mitochondrial Ca^2+^ measurements, EG7 cells were transduced with 100 × 10^3^ MOI units of viral particles (serotype 8) in the presence of 8 µg/mL polybrene (Sigma-Aldrich, St. Louis, MO, USA, # TR-1003) 48 h prior to the siRNA transfections as above so that measurements were conducted 4 d after viral transduction.

### 4.10. Westerm Blot

For protein expression analysis, cells were harvested, washed with PBS, then lysed in buffer containing 20 mM Tris, 100 mM NaCl, 10% glycerine (*v*/*v*), and 1% digitonin, pH 7.4. Lysates were stored at −20 °C until analyzed. Standard SDS-PAGE was performed followed by electrotransfer to PVDF membranes. Immunoblots were probed with primary antibodies against MCU (Cell Signaling, Danvers, MA, USA, # 14997, used at 1:500 dilution) or the housekeeping protein calnexin (ENZO, Lörrach, Germany, ADI-SPA-865, used at 1:1000 dilution) and the secondary HRP-coupled anti-rabbit (GE Healthcare, Munich, Germany, # NA9340V, used at 1:10,000 dilution). For protein detection, an enhanced chemiluminescence detection reagent was used (Clarity Western ECL Substrate, Biorad, Hercules, CA, USA). Densitometric quantification of detected protein bands was performed with Quantity one software (Biorad, Hercules, CA, USA). 

### 4.11. Statistical Analysis

For analysis of Ca^2+^ imaging data, 100–250 cells were measured per experiment, and 1–2 independent experiments were performed per mouse or transfection for 4–5 independent replicates. Otherwise, data are presented as mean ± S.E.M. Data were tested to determine whether they were normally distributed. When comparing two groups, statistical significance was tested by performing the unpaired, two-tailed Student’s *t*-test for normally distributed data sets and the Mann–Whitney test when samples were not normally distributed or when the sample size was not sufficient to test for normality. Asterisks indicate significant differences for different *p*-values as follows: * *p* < 0.05, ** *p* < 0.01, and *** *p* < 0.001. Statistical analysis was performed using Graphpad Prism.

## Figures and Tables

**Figure 1 ijms-24-07772-f001:**
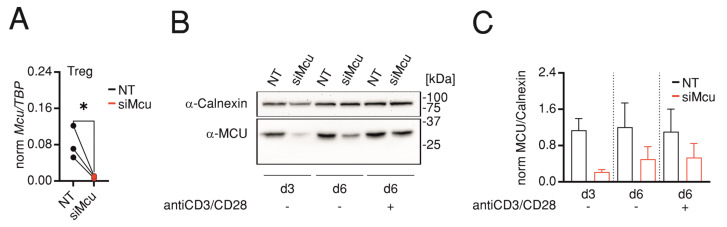
si*Mcu* treatment sustains low protein levels during the course of suppression assay. (**A**) qRT-PCR analysis showing normalized *Mcu* levels 3 days after transfection of murine regulatory CD4 T cells transfected with non-targeting (NT, black) or with *Mcu*-silencing (si*Mcu*, red) RNA. Results represent averages ± S.E.M of 3 independent experiments, * *p* < 0.05; (**B**) Representative Western blot of 3 independent transfections and corresponding quantification (**C**) of MCU protein levels analyzed in murine T cells (EG7) transfected and treated as in (A) but analyzed at Day 3 (d3) and Day 6 (d6) with our without TCR stimulation using anti-CD3/CD28-coated beads.

**Figure 2 ijms-24-07772-f002:**
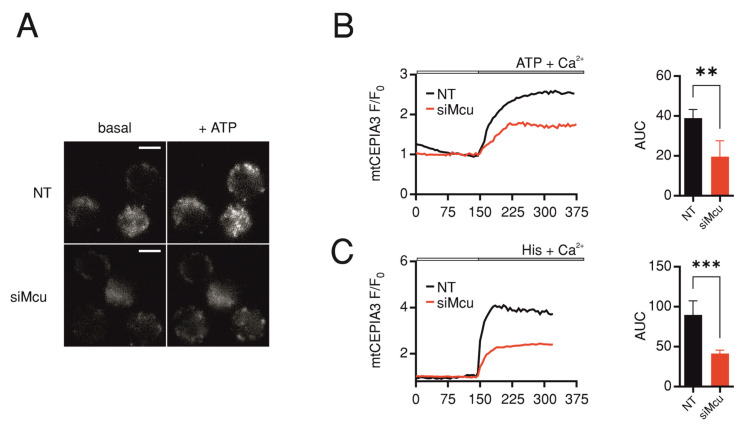
*Mcu* downregulation reduces mitochondrial Ca^2+^ uptake. (**A**) Representative images showing fluorescence of mtCEPIA3 corresponding to mitochondrial Ca^2+^ levels before (basal) and after application of 2 mM ATP in EG7 cells transfected with non-targeting (NT) or *Mcu*-silencing (si*Mcu*) RNA, scale bar represents 5 µm; Average traces showing changes of fluorescence ratio of mtCEPIA3 over time in cells transfected as in (**A**) where the mitochondrial Ca^2+^ uptake is triggered by application of 2 mM ATP (**B**) or 100 µM Histamine (**C**). Bar graphs represent averages ± S.E.M of area under influx curve (AUC) from 3 independent experiments, with 32–48 cells per condition. Results represent averages ± S.E.M of 3–5 independent experiments, ** *p* < 0.01, *** *p* < 0.001.

**Figure 3 ijms-24-07772-f003:**
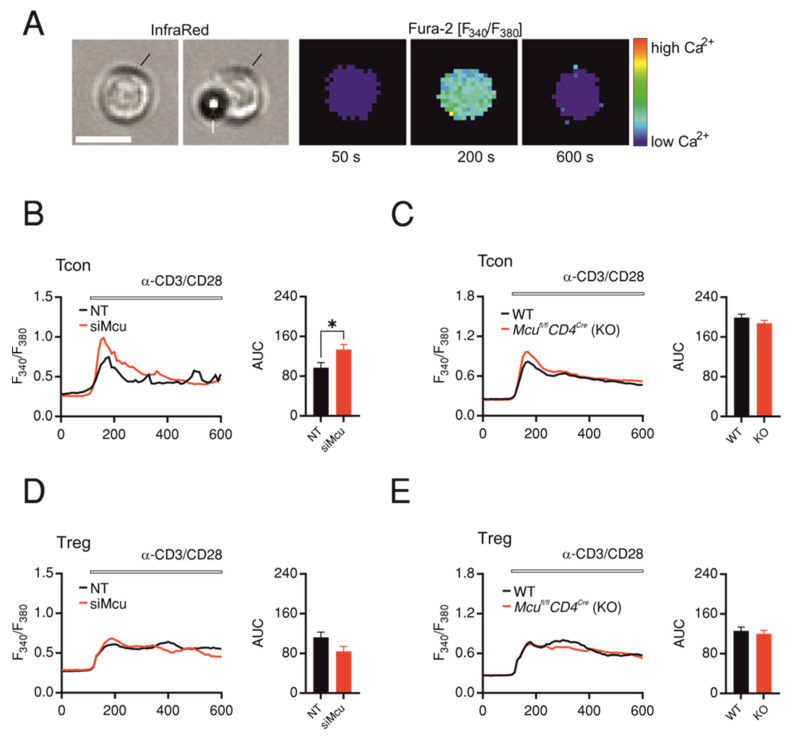
Transient downregulation of *Mcu* enhances SOCE in Tcons but not in Tregs. (**A**) Representative infra-red images of murine CD4 T cell before and after contact with anti-CD3/CD28-coated beads and fluorescent images showing changes of Fura-2 fluorescence ratio at 340/380 nm of the same cell before (50 s) and after TCR stimulation triggered by bead contact (200 and 600 s), scale bar represents 10 µm. Representative traces (left panels) of kinetics of Fura-2 fluorescence ratio over time upon contact with anti-CD3/CD28-coated beads in WT Tcons (**B**) or Tregs (**C**) transfected with non-targeting (NT, black) or *Mcu*-silencing (si*Mcu*, red) RNA. (**D**,**E**) Corresponding SOCE measurements in Tcons (**D**) or Tregs (**E**) derived from WT (*CD4*^Cre^, black) or from *Mcu*-knockout mice (*Mcu*^fl/fl^*CD4*^Cre^, red). The bar graphs (right panels in (**B**–**D**)) represent average ± S.E.M from 4–5 independent experiments (transfections or mice) and 100 to 250 cells. Results represent averages ± S.E.M of 3–5 independent experiments, * *p* < 0.01.

**Figure 4 ijms-24-07772-f004:**
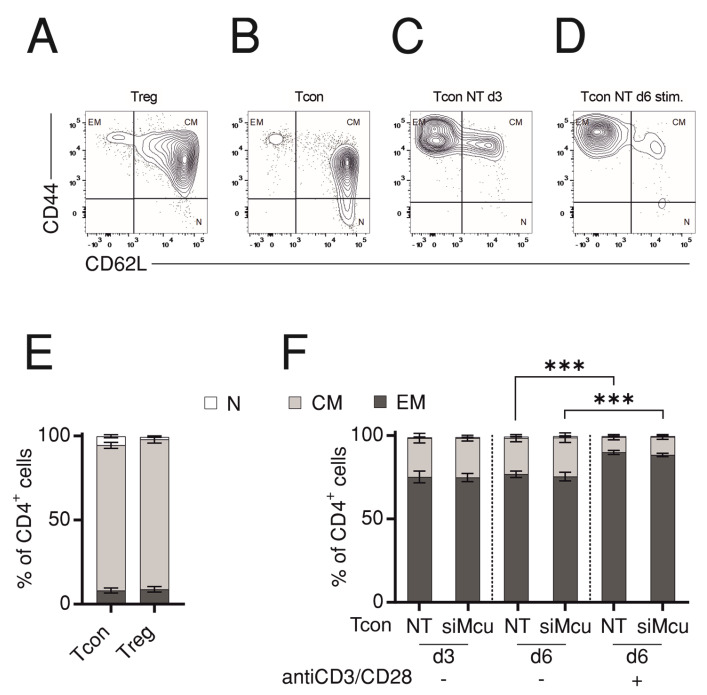
Downregulation of *Mcu* does not alter T cell fate decision. Representative flow cytometry plots showing compartment distribution in Tregs (**A**) or Tcons (**B**) after isolation or in Tcons at d3 (**C**) and d6 (**D**) of transfection with non-targeting (NT) or *Mcu*-silencing (si*Mcu*) RNA; (**E**,**F**) Bar graphs showing average fraction of naïve (N), central memory (CM), or effector memory (EM) compartments from cells analyzed as in (**A**–**D**) for surface expression of CD44 and CD62L. Results represent averages ± S.E.M from 3–4 independent experiments, *** *p* < 0.001.

**Figure 5 ijms-24-07772-f005:**
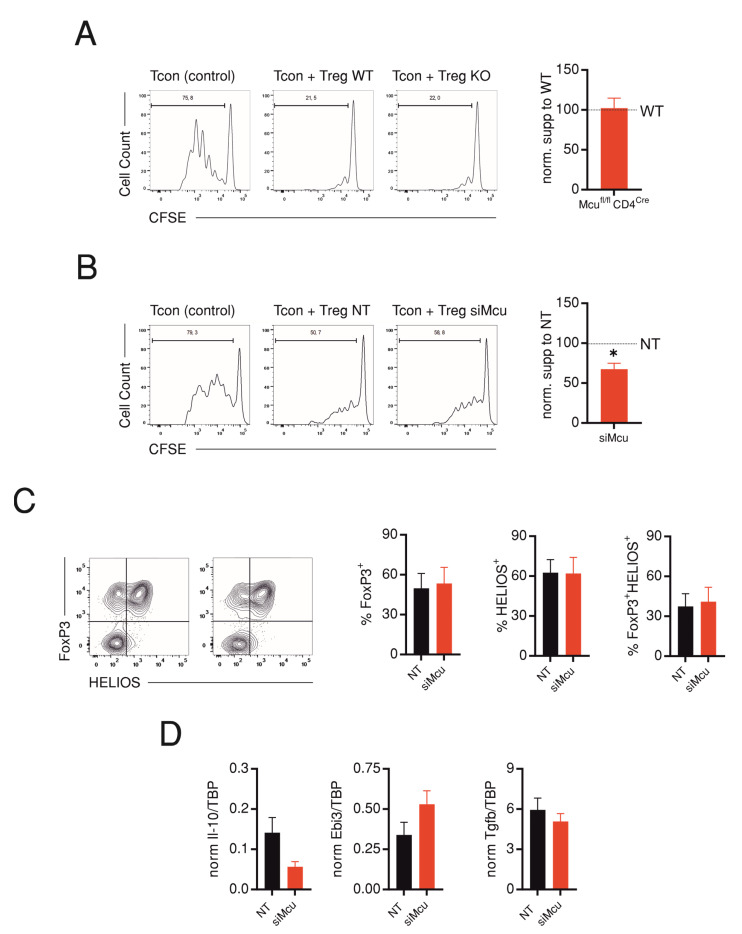
Transient downregulation but not genetic ablation of *Mcu* reduces suppressive capacity of Tregs. (**A**) Representative flow cytometric histograms showing proliferation of CFSE-labelled Tcons in absence (Tcon, control) or presence of Tregs derived from CD4^Cre^ (WT) (Tcon and Treg WT) or *Mcu*-KO (*Mcu*^fl/fl^*CD4*^Cre^) mice (Tcon and Treg KO). Right panel shows suppressive capacity of KO-Tregs normalized to the corresponding WT-Tregs (*n* = 5). (**B**) Analysis of suppressive capacity similar to (**A**) using Tregs derived from WT mice but transfected with non-targeting (NT) or *Mcu*-silencing (si*Mcu*) RNA (*n* = 6). (**C**) Representative flow cytometry plots showing expression of transcription factors FoxP3 and HELIOS in Tregs transfected as in (**B**). Right panel shows corresponding quantification from 3 independent experiments. (**D**) qRT-PCR analysis showing normalized cytokine expression levels: IL-10 (left panel), Ebi3 (middle panel), and TGF-β (right panel) in Tregs transfected as in (**B**) at Day 3 after transfection (*n* = 3). Results represent averages ± S.E.M from 3–6 independent experiments, * *p* < 0.05.

## Data Availability

All data presented in this study are described in the manuscript or the Appendix A and are available upon request through contacting the corresponding author.

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
