# Peer review of "Acute Downregulation but Not Genetic Ablation of Murine MCU Impairs Suppressive Capacity of Regulatory CD4 T Cells"

_ijms, 2023, doi:10.3390/ijms24097772_

Round 1
Reviewer 1 Report
In this manuscript, the authors show that acute inhibition of MCU using siMcu has different effects on cells when compared to chronic elimination of the gene in specific cell types. Furthermore inhibition of Treg suppressive activity is noted in acute treatment while it has been reported that there are no changes in Tregs in a model where the gene has been deleted in those cells. This data is surprising and raises a number of questions
1. How efficient was the introduction of siMcu in the different cell types. What % of cells had the control and siRNAs?
2. Do the siRNA transfected cells express any immunosuppressive cytokines.
3. Was there any sex difference in the data?
4. What percentage decrease in MCU protein was observed?
5. Was their any suggestion of IFN production due to the siRNA. For example was any increase in pSTAT1 observed?
6. Did the siMcu cause any cell death?
7. In Figure 1, how do the authors know that siMcu was still present in the cells at day 6? Without knowing this it is difficult to understand the interpretation that CD3/CD28 upregulates expression and abrogates the effects of the siRNA.
Reviewer 2 Report
In the manuscript „Acute downregulation but not genetic ablation of murine MCU impairs suppressive capacity of regulatory CD4 T cells“ Jost et al. show how siRNA-mediated downregulation (KD) of Mcu in murine T cells influences both mitochondrial and cytosolic Ca2+ uptake as well as suppressive capacity of Tregs. It has been previously shown that the MCU mediates mitochondrial Ca2+ uptake and that the genetic ablation (KO) of Mcu in mice models do not impair T cell-mediated immunity. On the other hand, in here presented manuscript authors report different effects when using siRNA-mediated downregulation of Mcu. Although the topic of the manuscript is very interesting, my biggest concern is whether here presented results are artefacts of the model/method used. It seems that here presented results could be due to off-target effects of siRNA or some compensatory mechanism of residual levels of Mcu after siRNA-mediated downregulation. For example, authors report increased expression of Mcu after TCR stimulation, however the effect of TCR stimulation was mostly observed in KD samples and not in controls, so one would expect that the TCR stimulation could influence the KD efficiency.
If the TCR stimulation influences/suppresses the efficiency of Mcu silencing and counteracts the siRNA-mediated Mcu downregulation, could authors explain quite unexpected results of enhanced SOCE upon TCR stimulation in Mcu KD T cells? One would expect that the due to suppression of Mcu downregulation via TCR stimulation the effects of Mcu in T cells would be negligible. Furthermore, the authors reported that the Mcu KD impairs the suppressive capacity of T regs during the course of TCR stimulation (e.g. after 3 days). However, authors reported that TCR stimulation influences Mcu KD efficiency after 6 days. Thus, it would be interesting to check the effect of TCR stimulation on efficiency of Mcu silencing in the course of TCR stimulation by WB, e.g. including several shorter timepoints.
In Figure 3, the AUC levels for NT are quite low (B) when compared to AUC levels for WT (C) T-con cells. Could authors please explain such differences in AUC levels between the control samples?
Although most of here presented results are obtained in vitro, the authors also used mice models, however the statement that the procedures on mice where approved by local animal protection committee and the local authorities are missing.
In their study, authors used mice models to perform genetic ablation of Mcu in T cells. However, the efficiency of Mcu KO by CD4 regulated Cre recombinase was not reported either on RNA or protein level.
Since no effect on FoxP3 expression was observed upon Mcu KD in Tregs, if possible, I would recommend the authors to include additional tests to confirm the suppressive capacity of Tregs upon Mcu-downregulation.
Some information in the Material and Methods section are missing, such as detailed information about antibodies used in WB (e.g., catalogue number and dilution for primary antibodies). In addition, please add sequences of forward and reverse primers for target and reference genes used in the qRT-PCR.
Minor points:
The authors explain that the TCR stimulation counteracts the siRNA-mediated Mcu KD. However, the description in the text refers to the effect after 3 days (rows 99-101), while the Figure 1B shows the effect of TCR stimulation after 6 days. Could authors please explain these discrepancies? To make the presentation of the results more clear, I recommend the authors to replace "d3" and "d6" with the "day 3" and "day 6", respectively, throughout the text. The designations d3 and d6 in figures can remain but should be described in the figure description.
From rows 179-184 authors are describing the results of Figure 5, however are referring to Figure 4.
There are some language misspellings throughout the manuscript, so I would recommend the authors to carefully read the paper and revise/rephrase the sentences as some are a bit unclear. Here are some examples that also include whole sentences:
Row 55 please leave out „in“ in „until in 2021”
Row 56 please replace „enhances” with “enhanced”
Rows 92-94: "To compare the potential adaptive effects resulting from genetic ablation to the transient downregulation of Mcu on suppressive capacity, we transfected murine Tregs with Mcu-specific siRNA."
Rows 94-96: "Analysis of Mcu-mRNA 72 h after transfection showed an efficient downregulation of Mcu in siRNA transfected cells to 11.5 ± 1.4% of Mcu levels in control cells transfected with non-targeting RNA (NT)."
Rows 115-118: "Measurements in two independent murine T cell lines (Fig. 2A and S1A) show that siMcu resulted in significant reduction of ATP and of histamine-induced mitochondrial Ca2+ uptake in agreement with the efficient downregulation we observed (Fig. 1) and with our previous findings using murine T cells with CD4 specific deletion of Mcu (Mcufl/flCd4Cre)"
Reviewer 3 Report
In the article "Acute downregulation but not genetic ablation of murine MCU impairs suppressive capacity of regulatory CD4 T cells" Jost and collegues showed interesting new insight in the role of MCU in Tregs.
The study sounds very interesting, however, there are some points to address.
Comments:
General point: Did the authors test for toxicity after the down-regulation of MCU.
In Fig 1 B the authors claim that this is a representative Western blot, of 4 independent transfections. However in the original Material are only 2 blots show and the Exp. A is different form Exp. B- they are not showing the same response. This might explain why the authors do not get statistical significance in Figure 1C. The authors should comment on this and provide information on the statistical test they used.
Under Point 2.3. "Mcu is dispensable for T cell fate decision" the authors described Fig. 4, however they did not go into detail, they should describe the sub-figures (4a-f).
Chapter "2.4. Downregulation of Mcu reduces suppressive capacity of Tregs" the authors describe Figure 4 which is supposed to be Figure 5!
Figure 5. Mcu-ko is most likely CD4-Cre x Mcufl/fl, just for clarification the authors should mention this in the legend. What do they use as control? CD4-Cre. How many experiments have been conducted in Fig 5. A and B. This not clear from the legend (most likely n=3 is true for all of the figures?)
If available, the authors should provide the licence number for their mouse experiments.
Round 2
Reviewer 1 Report
no further comments
Author Response
We thank the reviewer very much for the efficient reviewing process.
Best regards,
Dalia Alansary
Reviewer 2 Report
I would like to thank the authors for their answers and for improving their manuscript. In the revised version of the manuscript, the authors altered Figure 1 based on their findings, however the title of the Figure 1 should correspondingly be modified. I believe that the manuscript in the revised form meets the standard of IJMS and would recommend it for the publication with the above mentioned minor revision that refers to the requested changes in the title of the Figure 1.
Author Response
We thank the reviewer very much for the positive feedback and for accurate revision. We changed the title of Figure 1 to match the findings.
Best Regards,
Dalia Alansary
Reviewer 3 Report
The authors answered all my questions and improved the quality of the manuscript.
Author Response
We thank the reviewer very much for the efficient reviewing
Best regards,
Dalia Alansary